# Hole Injection Effect and Dynamic Characteristic Analysis of Normally Off p-GaN HEMT with AlGaN Cap Layer on Low-Resistivity SiC Substrate

**DOI:** 10.3390/mi13050807

**Published:** 2022-05-22

**Authors:** Hsiang-Chun Wang, Chia-Hao Liu, Chong-Rong Huang, Hsien-Chin Chiu, Hsuan-Ling Kao, Xinke Liu

**Affiliations:** 1College of Materials Science and Engineering, Shenzhen University–Hanshan Normal University Postdoctoral Workstation, Shenzhen University, Shenzhen 518060, China; smallflgt@hotmail.com; 2Key Laboratory of Optoelectronic Devices and Systems, Ministry of Education and Guangdong Province, College of Physics and Optoelectronic Engineering, Shenzhen University, Shenzhen 518060, China; xkliu@szu.edu.cn; 3Department of Electronic Engineering, Chang Gung University, Taoyuan 333, Taiwan; r3287133@gmail.com (C.-H.L.); gain525252@gmail.com (C.-R.H.); snoopy@mail.cgu.edu.tw (H.-L.K.); 4Department of Radiation Oncology, Chang Gung Memorial Hospital, Taoyuan 333, Taiwan

**Keywords:** p-GaN E-mode HEMT, normally-off, dynamic Ron, hole injection, V_TH_ shifting

## Abstract

A p-GaN HEMT with an AlGaN cap layer was grown on a low resistance SiC substrate. The AlGaN cap layer had a wide band gap which can effectively suppress hole injection and improve gate reliability. In addition, we selected a 0° angle and low resistance SiC substrate which not only substantially reduced the number of lattice dislocation defects caused by the heterogeneous junction but also greatly reduced the overall cost. The device exhibited a favorable gate voltage swing of 18.5 V (@I_GS_ = 1 mA/mm) and an off-state breakdown voltage of 763 V. The device dynamic characteristics and hole injection behavior were analyzed using a pulse measurement system, and Ron was found to increase and V_TH_ to shift under the gate lag effect.

## 1. Introduction

GaN power transistors have become key devices in high-power and high-efficiency power conversion systems, mainly because of their material properties, such as a wide band gap, high mobility, and strong electric breakdown field. Various approaches, such as the gate recessed structure [1,2,3], fluorine ion treatment [4], and a p-type GaN cap layer have been reported for giving these devices normally-off operation [5,6,7]. The other advantages of GaN power transistors are their high breakdown voltage, high switching speed, and low on-resistance. Therefore, p-GaN gate normally-off power devices have been considered as key devices in high power and high-frequency applications such as power conversion systems in hybrid or electric vehicles [8]. Normally-off operation is necessary for such applications because the current must be cut off in case of uncontrollable situations, such as short and open modes.

In commercial p-GaN HEMTs, device gates are either ohmic contacts or Schottky contacts. Compared with ohmic gates, such as gate injection transistors, Schottky gates have a lower forward gate leakage current, mainly due to a reverse Schottky diode at the junction of the metal and p-GaN. A Schottky gate p-GaN HEMT also exhibits a time-dependent gate breakdown voltage, but the gate’s operating voltage is limited to 6–7 V. Therefore, many research groups are developing methods to increase this maximum value so that the device can be operated in a wider gate bias range [9]. Under positive bias, the gate breakdown of a p-GaN HEMT can be attributed to the strong electric field concentrated at the metal/p-GaN interface [10]. Different from the time-dependent dielectric breakdown performance of Si and SiC power MOSFETs, the time-dependent gate breakdown behavior of p-GaN HEMTs is usually positively correlated with the temperature coefficient which means that high-energy carriers are accelerated through impact ionization or hot electron bombardment in a strong electric field. Therefore, one of the methods for improving the reliability of a gate is to make structural or process changes at the metal/p-GaN junction. Conversely, many scholars have analyzed the physical mechanism of dynamic resistance change which is mainly due to hot electron injection on the surface and defects in the buffer layer. These defects have numerous forms, such as carrier vacancies, lattice dislocations, and impurities. Additionally, GaN devices are mostly operated under high frequency and power. Thus, the characteristics of the device during high-temperature operation are also critical. Two substrates are employed in p-GaN HEMTs, namely GaN-on-Si and GaN-on-SiC. Compared with a GaN-on-Si HEMT, a GaN-on-SiC HEMT should be a more favorable choice for high-power switching components because of its high thermal conductivity, low resistivity, and high-voltage capability. Another advantage of using a SiC substrate is its lower lattice mismatch of approximately 3% for GaN (that of Si is ~17%). Therefore, a low-resistance and 0°-angle SiC substrate not only has the aforementioned advantages of GaN-on-SiC but also has a lower cost than a high-resistivity SiC substrate. We previously employed an AlGaN cap layer and low-resistivity SiC substrate in a p-GaN HEMT [11,12].

## 2. Device Structure

In this study, an AlGaN/p-GaN/AlN/AlGaN/GaN HEMT was grown on a 6-inch low-resistivity SiC substrate through metalorganic chemical vapor deposition. The epitaxial structure is illustrated in Figure 1a. An undoped GaN channel layer with a thickness of 300 nm was grown on an undoped AlGaN/GaN buffer/transition layer with a thickness of 4 μm. Subsequently, an Al_0.25_Ga_0.75_N barrier layer with a thickness of 15 nm and a p-type GaN layer with a thickness of 100 nm were grown.

Finally, an Al_0.2_Ga_0.8_N layer of 10 nm thickness was grown on the p-GaN layer. In the device fabrication, the p-GaN etching of Cl_2_/BC_l3_/SF_6_ was achieved using inductively coupled plasma, and the AlN layer acted as an etching stop layer. The etching stop technique employed was similar to that used in the p-GaN etching process [13,14,15]. The etching rate for p-GaN layer etching was approximately 31.5 nm/min. From 210 to 270 s, the etching rate was reduced to less than 2 nm/min because of generation of the AlF_3_ layer (etching stop layer). This etching stop technique prevents overetching in the p-GaN removal process, as shown in Figure 1b. Ohmic contacts were prepared through electron beam evaporation, and Ti, Al, Ni, and Au layers (thickness = 25, 120, 25, and 150 nm, respectively) were stacked on the device sequentially. Both devices were then annealed using a rapid thermal annealing system at 875 °C for 30 s in ambient N_2_. Finally, a Ni/Au (25/120 nm) gate metal stack was deposited and a 100-nm SiO_2_ passivation layer was applied.

## 3. Experimental Result and Discussion

Figure 2a,b reveal the log-scale transfer (I_DS_–V_GS_) and output (I_DS_–V_DS_) characteristics of the device. As illustrated in Figure 2a, the off-state current was 5 × 10^−5^ mA/mm at V_GS_ = 0 V. Additionally, the threshold voltage V_TH_ was 1.5 V which is defined at I_DS_ = 1 mA/mm. The corresponding maximum drain current density I_Dmax_ and Ron were 210 mA/mm and 20 Ω mm, respectively.

To observe the hole injection effect during gate operation, I_GS_–V_GS_ measurements were made [Figure 3a]. The device exhibited a large gate operation voltage under forward bias, and the gate turn-on voltage V_GS_ON_ was 18.5 V at I_GS_ = 1 mA/mm. The device exhibited favorable gate behavior because of its higher barrier which effectively suppressed the carrier injection. As revealed by Figure 3b, the device exhibited a high off-state breakdown voltage of 763 V. We used pulse measurement to analyze the hole injection effect and dynamic characteristics of the device under various stress voltages V_GSQ_ and durations. To evaluate the gate lag behavior we employed the AM-241 pulse measurement system [16,17]. The operation condition and gate lag measurement are illustrated in Figure 4a,b, respectively. Two bias conditions had to be considered: pulse voltage (V_GSP_ and V_DSP_) and quiescent voltage (V_GSQ_ and V_DSQ_). During the measurement, the pulse voltage switched rapidly to the quiescent voltage with a 2 µs pulse width and 200 µs period, and V_GSQ_ was swept from 0 to −15 V in increments of −5 V. The device exhibited dynamic Ron of 1.22 times at V_GSQ_ = −15 V.

As indicated in Figure 5a, V_TH_ shifted in the positive direction under V_GSQ_ = 6 V when the stress duration was increased from 0.1 ms to 1, 10, and 20 ms. This was caused by an enhanced electron injection from the channel at a higher V_GSQ_ and trapping at the p-GaN/AlN/AlGaN interfaces. Moreover, the V_TH_ shifting could be plotted in the pulse measurement, as presented in Figure 5b. When V_GSQ_ from 1 to 5 V was applied, holes may have accumulated at the p-GaN/AlGaN interface [Figure 6a] or in trap states at the AlGaN/GaN interface, temporarily increasing the density of the two-dimensional electron gas and causing a negative shift in V_TH_. When V_GSQ_ from 5 to 15 V was applied, V_TH_ shifted in the positive direction. This can be explained by some injected electrons being captured by the electron traps at the p-GaN/AlGaN interface, with the trapped electrons not being able to immediately escape [Figure 6b]. Subsequently, the stress voltage was larger than 15 V, and the hole injection was turned on for injection of the p-GaN/AlGaN interface, leading to recombination of the trapped electrons [Figure 6c]. Thus, the V_TH_ shift was reversed again [18,19,20,21,22].

To analyze the thermal characteristics of the device, its I_DS_–V_GS_ characteristic was measured under 25 °C to 175 °C with a 50 °C step [Figure 7a]. The device exhibited a V_TH_ shift of less than 0.3 V and Ron increased to 1.28× at 175 °C [Figure 7b]. Therefore, the p-GaN HEMT on a SiC substrate has high thermal stability because of its high thermal dissipation ability. Finally, the distributions of V_TH_ and Ron characteristics for 30 devices were measured and are presented in Figure 8. The mean V_TH_ and Ron were 1.5 V and 20 Ω·mm, respectively. Finally, the comparison of gate breakdown voltage and gate leakage current at V_GS_ = 6 V with each group were shown in Figure 9 [11,17,19,20,23,24,25,26].

## 4. Conclusions

In the device developed in this study, an AlGaN cap on the p-GaN layer reduced the hole injection effect and created a large gate operation range. It is hoped that the gate driver of traditional silicon devices can be shared and the operating safety voltage of the gate can be increased simultaneously. In addition, we grew a p-GaN HEMT with an AlGaN cap layer on a low-resistance SiC substrate, thus lowering the lattice defect density at the buffer layer position of the GaN-on-SiC structure. This improved the heat dissipation performance and lowered the cost of device production.

## Figures and Tables

**Figure 1 micromachines-13-00807-f001:**
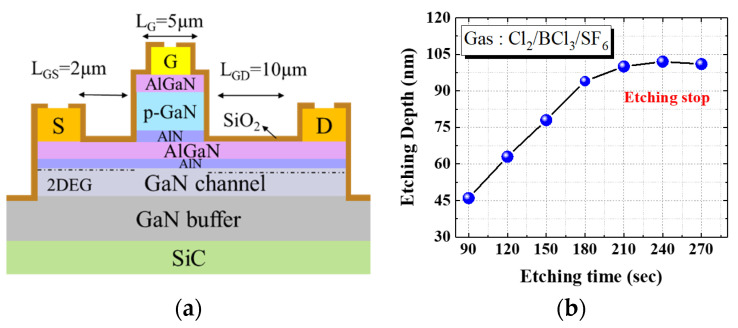
(**a**) Cross-sectional schematic of the p-GaN gate HEMT, and (**b**) the etching stop technique.

**Figure 2 micromachines-13-00807-f002:**
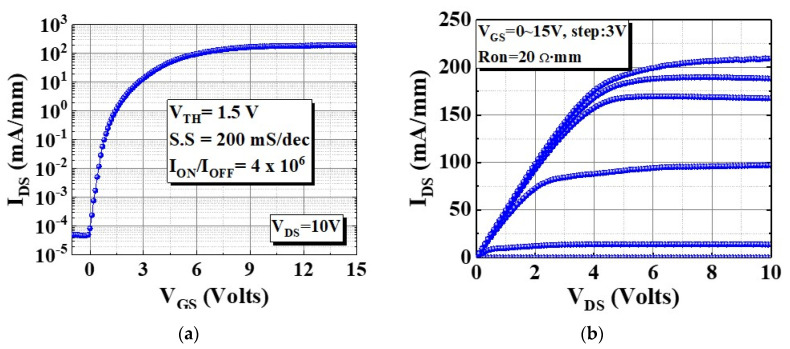
Device’s dc characteristics with L_GS_, L_G_, L_GD_, and W_G_ = 2, 5, 10, and 100 μm, respectively: (**a**) transfer and (**b**) output characteristics.

**Figure 3 micromachines-13-00807-f003:**
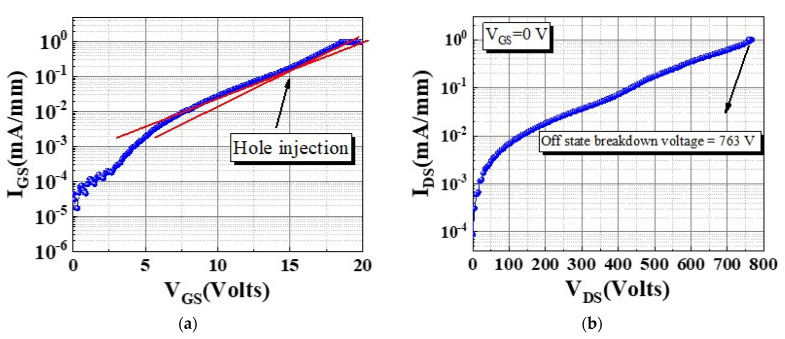
(**a**) I_GS_−V_GS_ characteristic and (**b**) off−state breakdown voltage measurement of the device.

**Figure 4 micromachines-13-00807-f004:**
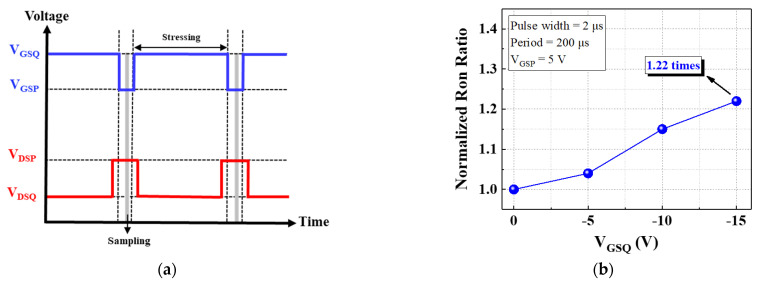
(**a**) Condition of pulse measurement and (**b**) gate lag characteristic.

**Figure 5 micromachines-13-00807-f005:**
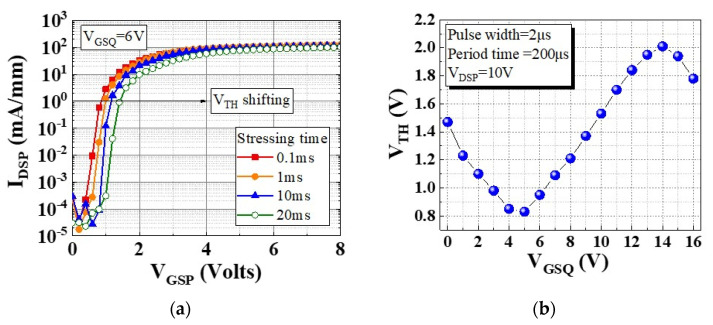
(**a**) Dynamic transfer characteristics measured for various stress durations, and (**b**) V_TH_ shifting for different V_GSQ_.

**Figure 6 micromachines-13-00807-f006:**
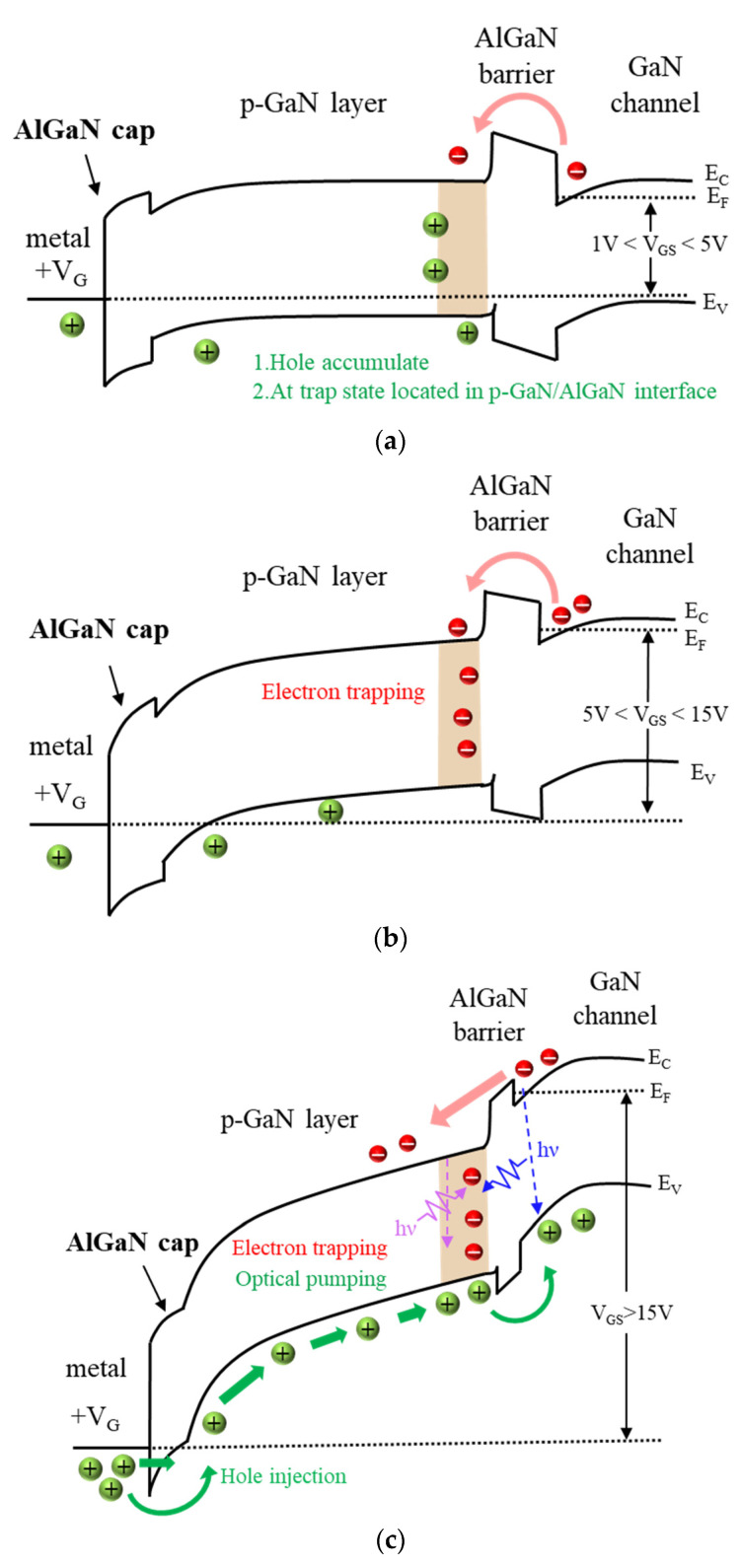
Schematic of the band diagram gate region of a p-GaN HEMT operating at (**a**) 1 V < V_GS_ < 5 V, (**b**) 5 V < V_GS_ < 15 V, and (**c**) V_GS_ > 15 V.

**Figure 7 micromachines-13-00807-f007:**
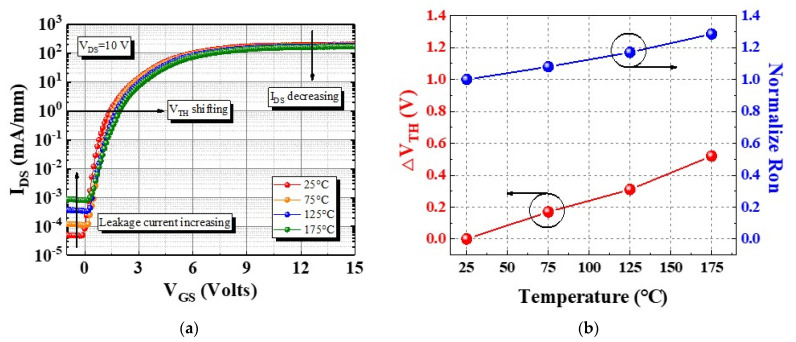
(**a**) I_DS_–V_GS_ characteristics of the device at 25 °C to 175 °C, and (**b**) variation in V_TH_ and Ron.

**Figure 8 micromachines-13-00807-f008:**
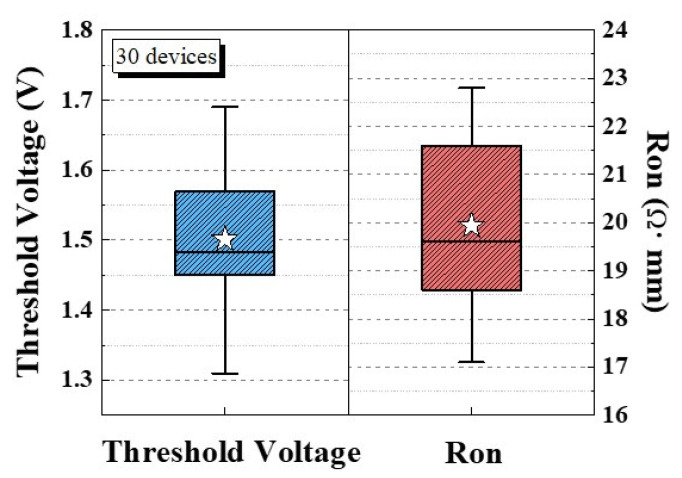
V_TH_ and Ron distributions for 30 devices.

**Figure 9 micromachines-13-00807-f009:**
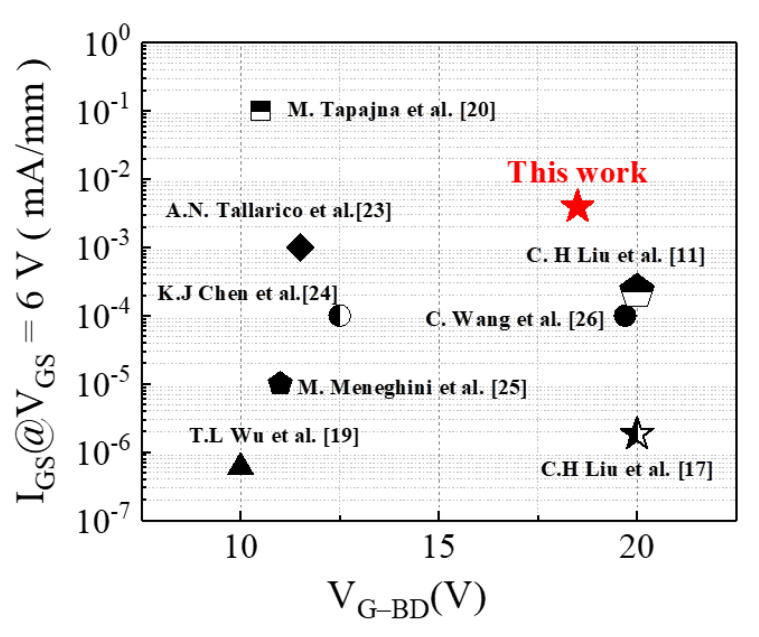
Comparison with other teams of the V_GB−D_ and I_GS−Leakage_.

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
