# Peer review of "Hole Injection Effect and Dynamic Characteristic Analysis of Normally Off p-GaN HEMT with AlGaN Cap Layer on Low-Resistivity SiC Substrate"

_micromachines, 2022, doi:10.3390/mi13050807_

Round 1
Reviewer 1 Report
The authors have reported hole injection effect and dynamic characteristics analysis of normally-off p-GaN HEMT. I have several questions/suggestions before I can recommend the manuscript for publication.
- Please discuss the significance of p-GaN HEMT devices -i.e., why we need this in more details in the introduction.
- Please add the relevant references in the introduction. For example: many research groups are developing … (need reference), gate breakdown of a p-GaN HEMT can be attributed to …(need reference).
- Correct the incomplete line in page 2, line 68.
- In pg. 3, line 85, how is the threshold voltage defined and in line 91, how is the gate turn-on voltage defined?
- How does the threshold voltage change as a function of stressing time for the negative gate quiescent bias?
- What is the drain quiescent bias condition for Fig. 5(a)?
- Please add more details about how the band-diagrams are calculated. What method/software is used; what materials parameters are used?
- Please add a comparison of the performance of the reported p-GaN device with the existing literature.
Author Response
Reviewer 1:
Q1. Please discuss the significance of p-GaN HEMT devices -i.e., why we need this in more details in the introduction.
Q2. Please add the relevant references in the introduction. For example: many research groups are developing … (need reference), gate breakdown of a p-GaN HEMT can be attributed to …(need reference).
Q3. Correct the incomplete line in page 2, line 68.
Q4. In pg. 3, line 85, how is the threshold voltage defined and in line 91, how is the gate turn-on voltage defined?
Q5. How does the threshold voltage change as a function of stressing time for the negative gate quiescent bias?
Q6. What is the drain quiescent bias condition for Fig. 5(a)?
Q7. Please add more details about how the band-diagrams are calculated. What method/software is used; what materials parameters are used?
Q6. Please add a comparison of the performance of the reported p-GaN device with the existing literature.
Ans:
- Thanks for the suggestion. “Therefore, p-GaN gate normally-off power devices have been considered as key devices in high power and high-frequency applications such as power conversion systems in hybrid or electric vehicles [8]. Normally-off operation is strongly required for such ap-plications because the current should be cut off in case of uncontrollable situations, such as short and open modes”, which were added in the revised manuscript.
- The relevant references are added in the manuscript and the incomplete line in page
2, line 68 was corrected.
- Thanks for your careful reading and the incaution mistake was corrected in the
revised version.
- The device threshold voltage VTH was 1.5 V which is defined at IDS=1mA/mm and
the gate turn-on voltage (VGS_ON) was 18.5 V at IGS = 1 mA/mm.
- In this work, the device focus on the ON-state operation so we just analyzed the VTH shifting at different positive VGSQ. However, at reverse gate bias with VGSQ < 0V, the Schottky diode is forward biased, with a decreased depletion width in the p-GaN layer and holes in the p-GaN layer are emitted to the gate metal, discharging the capacitor related to the reverse-biased heterojunction. When VGSQ is switched back to positive levels, the reduced positive charges in the p-GaN layer cannot be restored immediately, thus leading to a positive shift in the VTH [Ref.1].
Ref.1 : Hanxing Wang, JinWei, Ruiliang Xie, Cheng Liu, Gaofei Tang, and Kevin J. Chen, “Maximizing the Performance of 650-V p-GaN Gate HEMTs: Dynamic RON Characterization and Circuit Design Considerations”, IEEE TRANSACTIONS ON POWER ELECTRONICS, VOL. 32, NO. 7, JULY 2017.
- In Fig. 5(a), the drain quiescent bias is 0V.
- Our group always use TCAD to simulate the bandgap and electron field of the device under different bias as shown in Fig.1 in this reply file. However, to express the carrier transport behavior under different gate bias, so Fig. 6 is schematically illustrated about the processes of electron and hole injection/trapping and hole injection. The band-diagram is reference from two paper which are referenced in manuscript [17] and [20].
The comparison of gate breakdown voltage and gate leakage current at VGS = 6V with each group were shown in Fig. 2.

Reviewer 2 Report
The authors design a p-GaN gate stack with AlGaN cap layer. In addition, the GaN HEMT was fabricated on low-R SiC to enhance epitaxial quality and heat dissipation. All these are interesting studies for the researchers in this area. Neverthless, the devices in this study did not show very competitive Ron and breakdown voltages. Moreover, the characterization results of the devices with AlGaN cap on p-GaN gate or on SiC are relatively limited with insufficient depth. For example, the potential capability of using SiC substrate to obtain epitaxy with better quality and the resultant dynamic Ron should be discussed in more details.
Author Response
The authors design a p-GaN gate stack with AlGaN cap layer. In addition, the GaN HEMT was fabricated on low-R SiC to enhance epitaxial quality and heat dissipation. All these are interesting studies for the researchers in this area. Neverthless, the devices in this study did not show very competitive Ron and breakdown voltages. Moreover, the characterization results of the devices with AlGaN cap on p-GaN gate or on SiC are relatively limited with insufficient depth. For example, the potential capability of using SiC substrate to obtain epitaxy with better quality and the resultant dynamic Ron should be discussed in more details.
Ans: Thanks for your valuable suggestion.
The p-GaN gate stack with AlGaN cap layer on low-R SiC is designed which base on the experience of structure research in our group [10], [11]. So, some about performance have been discussed in previous studying. This studying more discussed at hole injection behavior, high temperature effect, and gate lag effect. Owing to the high material properties of gallium nitride and the SiC substrate, these devices are expected to operate in high-temperature environments. In addition, the p-GaN HEMT with AlGaN cap layer exhibit the high gate reliability, but it needs to trade-off between Ron and gate operation range. Due to the weak AlGaN cap layer induced worse gate-to-channel modulation ability, a higher VGS need to be applied on device to reach the saturation current. However, the device exhibits a high off-state breakdown voltage of 763 V which is higher than the specification in commercial p-GaN power device.
Moreover, the XRD was used to investigate the dislocation density, and the result are shown in this revision reply file. The rocking curve scan of a (002) reflection provides information on the degree of tilt with respect to the surface of a device, and the FWHM of this reflection is a qualitative measure of screw dislocation density (Nscrew). The rocking curve scan of a (102) reflection provides information on the degree of twist with respect to the surface of a device, and the FWHM of this reflection is a measure of edge dislocation density (Nedge). The dislocation density can be calculated using XRD-derived FWHM results as follows:
(2) |
where Nscrew and Nedge are the screw and edge dislocation densities, respectively, and is Burger’s vector. The full width at half-maximum value of the (002) asymmetric and (102) asymmetric reflection were used to measure crystal quality. As the results, GaN on low resistivity SiC (LR-SiC) substrate exhibit better dislocation density of 4.22108 cm-2 which indicate there is a better epi quality in GaN on LR-SiC substrate.

Round 2
Reviewer 1 Report
I would like to thank the authors for addressing my initial comments. I recommend the manuscript for publication.
Reviewer 2 Report
The authors have answered my comments.